# Resilience Against COVID-19: How Italy Faced the Pandemic in Pediatric Orthopedics and Traumatology

**DOI:** 10.3390/children8070530

**Published:** 2021-06-22

**Authors:** Giovanni Trisolino, Renato Maria Toniolo, Lorenza Marengo, Daniela Dibello, Pasquale Guida, Elena Panuccio, Andrea Evangelista, Stefano Stallone, Maria Lucia Sansò, Carlo Amati, Pier Francesco Costici, Silvio Boero, Pasquale Farsetti, Nando De Sanctis, Fabio Verdoni, Antonio Memeo, Cosimo Gigante

**Affiliations:** 1Unit of Pediatric Orthopaedics and Traumatology, Istituto Ortopedico Rizzoli, IRCCS, 40136 Bologna, Italy; stallone.stefano@gmail.com; 2Department of Orthopaedics and Traumatology, Bambino Gesù Children’s Hospital, IRCCS, 00146 Rome, Italy; renatomaria.toniolo@opbg.net; 3Unit of Pediatric Orthopaedics and Traumatology, Istituto Giannina Gaslini, IRCCS, 16147 Genova, Italy; lorenzamarengo@libero.it (L.M.); silvio.boero@fastwebnet.it (S.B.); 4Unit of Pediatric Orthopaedics and Traumatology Giovanni XXIII Children’s Hospital, University of Bari, 70126 Bari, Italy; daniela.dibello@policlinico.ba.it (D.D.); dr.carloamati@gmail.com (C.A.); 5Unit of Pediatric Orthopaedics and Traumatology, Azienda Ospedaliera di Rilievo Nazionale Santobono Pausillipon, 80122 Napoli, Italy; ple.guida@gmail.com (P.G.); marialuciasanso@gmail.com (M.L.S.); 6Department of Paediatric Orthopaedics and Traumatology, Centro Specialistico Ortopedico Traumatologico Gaetano Pini-CTO, 20122 Milano, Italy; elenapanuccio@gmail.com (E.P.); Antonio.Memeo@asst-pini-cto.it (A.M.); 7Unit of General Affairs, Istituto Ortopedico Rizzoli, IRCCS, 40136 Bologna, Italy; andrea.evangelista@ior.it; 8Orthopedic Unit, Department of Surgery, Bambino Gesù Children’s Hospital, IRCCS, 00050 Rome, Italy; pierfrancesco.costici@opbg.net; 9Department of Orthopaedics Surgery, University of Rome “Tor Vergata”, 00133 Rome, Italy; farsetti@uniroma2.it; 10Unit of Pediatric Orthopaedics and Traumatology, Campolongo Hospital, 84025 Marina di Eboli, Italy; nandodesanctis@gmail.com; 11Unit of Pediatric Orthopaedics and Traumatology, Istituto Galeazzi, IRCCS, 20161 Milan, Italy; verdonifabio@alice.it; 12Pediatric Orthopaedic Unit, Department of Woman and Child Health, Padua General Hospital, 35121 Padua, Italy; cosimo.gigante@libero.it

**Keywords:** COVID-19, pediatrics, orthopedics, traumatology, surgery, statistics

## Abstract

Background: We aimed to investigate the variation of medical and surgical activities in pediatric orthopedics in Italy, during the year of the COVID-19 pandemic, in comparison with data from the previous two years. The differences among the first wave, phase 2 and second wave were also analyzed. Methods: We conducted a retrospective multicenter study regarding the clinical and surgical activities in pediatric orthopedics during the pandemic and pre-pandemic period. The hospital databases of seven tertiary referral centers for pediatric orthopedics and traumatology were queried for events regarding pediatric orthopedic patients from 1 March 2018 to 28 February 2021. Surgical procedures were classified according to the “SITOP Priority Panel”. An additional classification in “high-priority” and “low-priority” surgery was also applied. Results: Overall, in 2020, we observed a significant drop in surgical volumes compared to the previous two years. The decrease was different across the different classes of priority, with “high-priority” surgery being less influenced. The decrease in emergency department visits was almost three-fold greater than the decrease in trauma surgery. During the second wave, a lower decline in surgical interventions and a noticeable resumption of “low-priority” surgery and outpatient visits were observed. Conclusion: Our study represents the first nationwide survey quantifying the impact of the COVID-19 pandemic on pediatric orthopedics and traumatology during the first and second wave.

## 1. Introduction

The year 2020 will be remembered as the “year of COVID-19” [1]. The pandemic has dramatically impacted the organization and order of priorities in healthcare services globally. Italy was the first country in the Western world to suffer from the pandemic, and one of the most severely affected [2].

Following the discovery of the first clusters in late February 2020, an initial lockdown involving sixteen million people in northern Italy was imposed by the Italian government, and further turned into a national hard lockdown on 9 March 2020. As the contagion rate and death toll decreased, the so-called “Phase 2” started, with progressive reopening of ordinary activities involving social interactions by the second half of May. During the summer, people were free to move through Italy and many asymptomatic subjects likely contributed to spreading the infection and precipitating a more severe second wave that unsurprisingly started from September. In autumn, the Italian Government provided new rules, introducing a three-tier system of progressive restrictions, based on the local coronavirus situation. From March 2020 to the end of February 2021, the number of positive cases in Italy exceeded 3 million, with over 100,000 deaths, in a country of about 60 million people [3].

During this year, we have observed a drastic decline in the so-called “non-essential” health services, in all fields of medicine. Although there is no consensus in the definition of “elective” or “non-essential” procedures, many orthopedic activities can be considered elective or non-essential [4]. This led to a dramatic reduction in orthopedic activities from March 2020 in Italy, as in other countries [5,6].

Even pediatric orthopedics had to face this state of emergency. The Italian Society of Pediatric Traumatology and Orthopedics (SITOP) immediately drafted, disseminated across the pediatric Italian orthopedic community, and published specific recommendations to postpone elective or non-essential activities, while ensuring the continuation of essential health services [7]. Many Italian referral centers for pediatric orthopedics and traumatology directly participated in drafting these recommendations and adopted their rules.

To date, no data have been published concerning the effects of the COVID-19-related restrictions in pediatric orthopedics and traumatology in Italy.

Therefore, we aimed to investigate the variation of medical and surgical activities in pediatric orthopedics in Italy, during the year of COVID-19, in comparison with the previous two years. We also analyzed the differences among the first wave, phase 2 and second wave.

## 2. Materials and Methods

We conducted a retrospective multicenter analysis regarding clinical and surgical activities in pediatric orthopedics during the pandemic and pre-pandemic period.

The study involved seven public institutions, all of them were regional tertiary referral centers for pediatric orthopedics, included in the list of the recommended hospitals of the SITOP, which consists of fifteen hospitals. Four of the participant centers were pediatric hospitals, two were orthopedic hospitals and one was a university general hospital. Four of them were in the north of Italy, while three were in the south. Moreover, to investigate if further specific features of the hospital setting, apart from the geographical location, could influence the results, we divided the centers into:

(1)“High-volume centers” (>100 surgeries per month during the period 2018–2019) and “low-volume centers” (<100 surgeries per month during the period 2018–2019).(2)“Predominant elective surgery” (<30% trauma surgery per year during the period 2018–2019) and “predominant trauma surgery” (>30% trauma surgery per year during the period 2018–2019).

According to these categories, three hospitals were high-volume centers and three hospitals predominantly performed trauma surgery.

The hospital databases of these centers were queried for events regarding pediatric orthopedic patients (inpatient, outpatient and Emergency Department (ED) admissions, surgical interventions) from 1 March 2018 to 28 February 2021. The data were reported as divided by quarter. We included children and adolescents aged less than 18 years requiring orthopedic management. Diagnoses and surgical procedures were assessed according to the ICD-9 codes system and registered in a dedicated database.

The surgical procedures were classified according to the “SITOP Priority Panel”. Briefly, this panel categorizes surgical procedures as “urgent” (interventions that should be performed in less than 72 h) and “elective” (interventions that can be safely performed in more than 72 h), the latter being further divided into four priority levels, based on their clinical condition: “Priority A” refers to those operations that should be performed within 30 days of waiting; “Priority B” refers to those operations that should be performed within 90 days of waiting; “Priority C” refers to those operations that should be performed within 180 days of waiting; “Priority D” refers to those operations that could be safely postponed for more than 6 months (see Table 1) [7].

Seven raters performed data extraction and agreement was achieved by consensus, in cases of surgical procedures that could not be easily included in a specific category. The classification of elective surgical operations according to the priority panel is reported in Table 1 [7]. Moreover, those interventions classified as priority A and B were considered as “high-priority” surgery, while those interventions classified as priority C and D were overall considered as “low-priority” surgery.

### Statistical Analysis

Percentages of relative change of activity across periods, with 95% confidence intervals (95% C.I.), were estimated using mixed effect Poisson models considering each center as a random effect. Heterogeneity by center on activity changes was investigated including a random slope for the period variable (2020 vs. 2018–2019) at the center level and significance was evaluated with the likelihood ratio comparing the models with those that included the random intercept only. For the purpose of the study, we analyzed the trends of ED visits, outpatient visits and surgical operations (taken as a whole and stratified by priority) during the study period. To evaluate the homogeneity of variation between high-priority and low-priority surgery, we conducted mixed effect Poisson regression models, including an interaction term between period and priority variables. The same approach was used to investigate the homogeneity of variation between the number of ED visits and trauma surgery, and between the number of outpatient visits and elective surgeries. A *p*-value < 0.05 was considered statistically significant. All data were analyzed using STATA 11.2. (StataCorp, College Station, TX, USA).

## 3. Results

The total number of surgical interventions was 8283 in 2018, 8628 in 2019 and 6453 in 2020. Surgical volumes increased slightly but significantly between 2018 and 2019, mainly due to a greater number of trauma surgeries (see Appendix A). Therefore, to avoid overestimation of the drop in medical and surgical volumes during the year of the pandemic, we compared the data of 2020 versus the pooled data of the whole period of 2018–2019.

Overall, in 2020, we observed a significant drop in surgical activities of −23.7% (Table 2). Analyzing the data by each center, we observed a significant heterogeneity among hospitals, due to geographical location (hospitals in northern Italy were more affected by the decrease in medical and surgical volumes) and to predominant surgical activity, with hospitals with predominant trauma surgery being less affected (Appendix A).

The decrease in surgical interventions followed the course of the contagion and the related restrictions, with the largest decline recorded during the first wave (−50.3%). However, although the surgical volumes significantly increased during phase 2, they never reached pre-pandemic levels; indeed, there was a further decline during the second wave (−19.7%), albeit much more contained than the first (Figure 1 and Table 3).

As expected, the decrease was different across the different classes of priority (Figure 2a), with trauma surgery being less influenced (−13.6%), while “low-priority” surgery suffered the highest burden of reduction (−29.8%) during the entire pandemic period. This difference was evident during the summer months when, besides the increase in pediatric trauma, likely related to the resumption of outdoor sports and recreational activities, it was necessary to reschedule the “high-priority” elective surgery backlog. Remarkably, during the latest quarter of the study period, we observed that “low-priority” surgery had a significant rebound, outpacing “high-priority” surgery (Figure 2b, Appendix A).

The overall number of outpatient visits was 68,569 in 2018, 71,837 in 2019 and 55,186 in 2020, with a 21.4% reduction compared to the average numbers of the pre-pandemic period (Table 2). Again, the highest relative reduction was recorded during the first quarter, with 56.8% of visits cancelled or postponed, while in the following months, there was a decrease of less than 10% per quarter, compared with the pre-pandemic period (Table 3). Moreover, the decrease in outpatient visits was slightly but significantly inferior to the decrease in elective surgery (−21.4% versus −27.9%; Figure 3a, Appendix A).

The number of ED visits was 33,074 in 2018, 32,816 in 2019, and 20,809 in 2020, decreasing by 39.1% during the pandemic period (Table 2). The decrease in ED visits was almost three-fold greater than the decrease in trauma surgery (−39.1% versus −13.6%; Figure 3b, Appendix A).

## 4. Discussion

The present study represents the first nationwide survey concerning the impact of the COVID-19 pandemic on pediatric orthopedics. We found that more than 25% of medical and surgical activities, corresponding to more than 15,000 visits and more than 2000 operations, were cancelled or postponed during this year. As expected, the centers in northern regions (the regions more severely affected by the pandemic) and the main institutions designed for pediatric orthopedic elective surgery suffered the most drastic decline.

We observed that every medical and surgical activity, even emergency and high-priority surgery, declined precipitously by more than 50% during the first wave. At that time, the national healthcare service was essentially unprepared to face the pandemic; a drastic reduction in all elective and non-essential treatments was the only strategy that could be adopted for allowing immediate reallocation of resources to counter the pandemic [2,5]. Thanks to the increased availability of tests and personal protective equipment, and due to the minimal infection numbers during the summer, many restrictions were revoked during the second and third quarter, and some non-essential medical and surgical activities resumed, although not at the same pre-pandemic levels. The national healthcare system remained prepared to face the predicted second wave, maintaining most of the strategies adopted to contain the contagion.

The second wave predictably came in autumn, more violent than the first. Nonetheless, the difference between the first and the second wave was evident. We reported an inferior decrease in surgical volumes, and outpatient visits. Furthermore, we observed a switch in volumes between “high-priority” and “low-priority” surgery, with the latter almost reaching pre-pandemic levels. This was probably due to the maintenance and implementation of some strategies identified for the post-peak period, in preparation of a second wave [7]. For instance, the coordination of local health facilities to redistribute outpatient services allowed the number of visits to increase, avoiding overcrowding of the hospital facilities. On the other side, the strict compliance with the priority class of intervention favored some typical pediatric orthopedic surgeries, such as minimally invasive or arthroscopic surgery, growth modulation surgery, and, in general, surgery that requires minimal hospitalization and no use of intensive care.

Our findings substantially confirm a better preparation of the Italian national health system for the more violent second wave. In contrast, the drop in numbers during the second wave was much higher, for example, in Germany [8].

In our study, trauma surgery covered almost 30% of the whole surgical volumes. We observed a significant decrease in trauma surgery, in line with several reports from other countries and from the adult counterpart [9,10,11,12,13,14,15]. We agree with some authors that this decline in emergency referrals was more related to the combination of seasonal trends of pediatric trauma and concomitant restriction of sportive and outdoor recreational activities during the lockdown periods, rather than the effect of contraction or reallocation of healthcare resources [10,11,12].

Remarkably, the decline of trauma surgery did not parallel the decrease in ED visits, with the latter having decreased much more consistently than the former. On the other side, we did not register a significant increase in delayed surgical treatments for neglected or mistreated injuries, as recently reported by Patel et al. [15].

A possible explanation of these findings is that parents and/or practitioners self-managed minor injuries, reducing inappropriate referrals to secondary care, in an attempt to avoid the risk of COVID-19 infection; on the other hand, pediatric orthopedic surgeons empowered some kind of telemedicine service, as recommended by the SITOP and by several orthopedic societies, to reduce the hospital admission as much as possible to only those cases that likely required urgent treatment [7,16,17]. Telemedicine has been largely and effectively used during the pandemic [10,17,18]. The advent of models such as virtual clinics is completely re-designing the concept of care, offering a viable, efficient approach to treat a variety of pediatric musculoskeletal disorders, but this needs further studies, especially regarding legal, safeguarding, safety and security issues. [18,19,20,21]. Another and more worrying explanation could be that, while the overall volumes of trauma patients decreased, the injury pattern changed, as reported in recent studies concerning adult trauma [13,14], thus requiring hospitalization more frequently. Lockdown- and quarantine-related measures, such as school closings and social isolation, could have increased domestic violence, child abuse or neglect, and self-harm. Unfortunately, our study was not designed to distinguish the type, causes and severity of trauma requiring surgery; therefore, we cannot provide further evidence in favor of or against this hypothesis. Nonetheless, we emphasize that outreach and injury prevention messaging may be useful in response to the potential adverse effects of lockdown or quarantine, especially in children [13,22,23].

While the reduction in trauma surgery can be partially explained by other causes, the reduction in elective surgery is mostly explainable by the contraction and reorganization of the healthcare services.

During this year, we have seen a drastic reduction of about a quarter in elective medical and surgical activities in pediatric orthopedics, albeit the reduction did not impact equally all priority classes. “Low-priority” surgery was more largely affected, with an overall annual loss of over 1000 operations. Nonetheless, elective surgery does not mean “optional surgery” [24]. Most of the elective surgery in pediatric orthopedics regards children with congenital malformations, neuromuscular disorders, and genetic and rare diseases. In this population, some interventions such as growth modulation, osteotomies, or soft tissue procedures may achieve the dual purpose of improving patient’s quality of life and reducing the need for more invasive operations at the end of growth. Moreover, “high-priority” elective surgery was reduced during the “year of COVID-19”. If the decrease in “priority B” interventions (which often concern the treatment of neonatal conditions, such as CDH or CTEV) can be partially explained by the decline in the Italian birth rate [25], the decline in “priority A” interventions, especially in operations for bone tumors or infections, during the first quarter, is more concerning. In such cases, the delay of surgery could worsen the prognosis; this “side effect” has already been reported in other studies, as a potential consequence of the lockdown measures [26,27,28].

In our experience, an effort was made to resume elective pediatric orthopedic surgery, as showed by the surgical volumes during the second wave. Unfortunately, since October 2020, we are observing a significant increase in infection prevalence among children, which could hamper surgical scheduling [29].

Our study has limitations. The method of classifying elective surgery by priority could have some issues. For instance, surgery for bone tumors includes from biopsies for benign lesions to resections for malignancies. Therefore, the consequences of the reduction in surgery in this class could be less harmful if only operations for benign lesions were postponed. The method of capturing data by hospital databases, using the ICD-9 codes system, cannot accurately distinguish between conditions that present different degrees of severity. Moreover, despite the SITOP providing recommendations for reorganizing pediatric orthopedic activities during the pandemic, and such recommendations likely being implemented by all the participant hospitals, we cannot assess the effectiveness of such recommendations, since we lack a control group that did not adopt the recommendations. Finally, we could not assess the rate of children or parents that became positive to COVID-19 tests immediately after hospitalization. This information would be of great interest to assess the safety and effectiveness of our protocols to resume clinical and surgical activities during a pandemic.

## 5. Conclusions

In conclusion, our study represents the first nationwide survey quantifying the impact of the COVID-19 pandemic on pediatric orthopedics during the first and second wave. These observations provide further evidence and considerations that should be thought about when developing contingencies for coping with a pandemic.

## Figures and Tables

**Figure 1 children-08-00530-f001:**
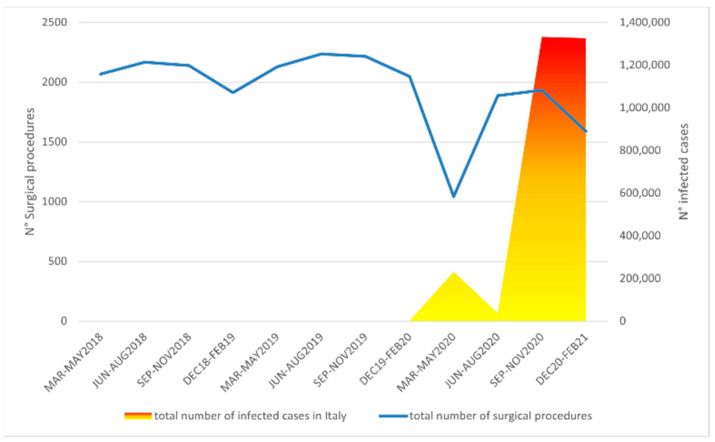
Graph illustrating the trend of surgical volumes in pediatric orthopedics during the period March 2018–February 2021, in relation to the pandemic situation in Italy.

**Figure 2 children-08-00530-f002:**
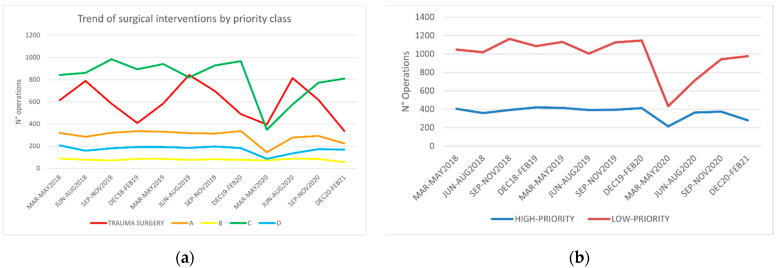
Trend of surgical operations during the period March 2018–February 2021: (**a**) divided by priority class; (**b**) divided high-priority/low-priority.

**Figure 3 children-08-00530-f003:**
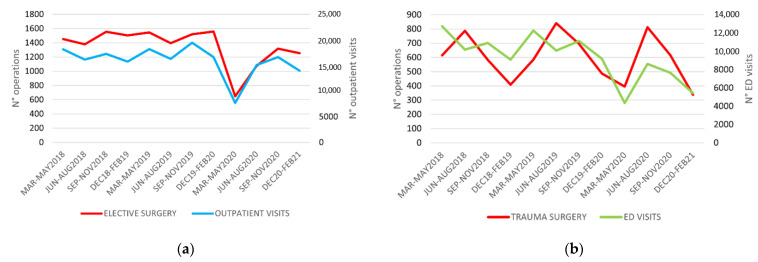
Trend of elective surgery during the period March 2018–February 2021: (**a**) in relationship with the outpatient visits; (**b**) in relationship with the Emergency Department (ED) visits.

**Table 1 children-08-00530-t001:** Priority class for elective pediatric orthopedic surgery [7]. OCD: osteochondral disease. CTEV: congenital talipes equinovarus. CDH: congenital dislocation of the hip. LCPD: Legg–Calve–Perthes Disease.

PRIORITY	A	B	C	D
TYPE OF SURGERY	Surgery for malignant or aggressive bone and soft tissue tumors.	Staple or guided growth hardware removal in case of overcorrection.	Minimally invasive surgery (percutaneous tenotomies, subtalar arthroereisis).	Surgical treatments in skeletally mature children.Arthrodesis.
	Biopsies for suspected malignancies.	Ponseti method for CTEV in older newborns (3–6 months).	Arthroscopic procedures.	Osteotomies of pelvis and long bones.
	Septic arthritis requiring arthroscopic lavage/sampling/evacuation.	Closed/open reduction and cast for CDH in older newborns (3–6 months).	Procedures that should be performed at a definite range of age (for example epiphysiodesis and hemi-epiphysiodesis at transitional age, treatments for congenital knee or foot and ankle dislocation, before start walking), or at a definite stage of disease (for example, osteotomies for LCPD).	Limb lengthening procedures.
	Slipped capital femoral epiphysis.			Spinal surgery for scoliosis.
	Misdiagnosed, neglected fractures or fractures displaced at follow-up.			
	Hardware-related complications (infection, migration).			
	Nerve injuries or compression with recent onset palsy not responding to nonoperative treatments.			
	Locked knee, bucket handle meniscal tear, loose bodies, OCD fragments.			

**Table 2 children-08-00530-t002:** Medical and surgical volumes in pediatric orthopedics and traumatology during the 2018–2020 period.

	YEAR			2020 vs. 2018–2019 Comparison ^1^		
	2018	2019	2020	Relative Change	95% C.I.	*p*-Value
**N° operations (total amount)**	8283	8628	6453	−23.7%	−25.8% to −21.5%	<0.001
***Trauma surgery***	2394	2608	2160	−13.6%	−17.9% to −9.2%	<0.001
***High Priority***	1576	1613	1232	−22.7%	−27.7% to −17.5%	<0.001
Priority A	1257	1295	936	−26.6%	−31.9% to −20.9%	<0.001
Priority B	319	318	296	−7.1%	−19% to 6.7%	0.298
***Low Priority***	4313	4407	3061	−29.8%	−32.6% to −26.8%	<0.001
Priority C	3577	3653	2502	−30.8%	−33.9% to −27.6%	<0.001
Priority D	736	754	559	−25%	−31.9% to −17.3%	<0.001
**Outpatient Visits**	68,569	71,837	55,186	−21.4%	−22.2% to −20.6%	<0.001
**ED visits**	42,954	42,649	26,071	−39.1%	−39.9% to −38.2%	<0.001

^1^ Comparison between the pandemic year (from 1 March 2020 to 28 February 2021) and the average numbers of the previous two years was performed using a mixed effects Poisson regression model with the center as the random effect. ED: Emergency Department.

**Table 3 children-08-00530-t003:** Relative change in medical and surgical volumes, during the year of COVID-19, divided by quarter.

	1 MAR–31 MAY		1 JUN–31 AUG		1 SEP–30 NOV		1 DEC–28 FEB	
	% Relative Change ^1^ (95% C.I.)	*p*-Value	% Relative Change ^1^ (95% C.I.)	*p*-Value	% Relative Change ^1^ (95% C.I.)	*p*-Value	% Relative Change ^1^ (95% C.I.)	*p*-Value
**N° operations** **(total amount)**	−50.3% (−53.6 to −46.9)	<0.001	−14.2% (−18.7 to −9.5)	<0.001	−11.2% (−15.8 to −6.3)	<0.001	−19.7% (−24.2 to −14.9)	<0.001
***High Priority***	−39.5% (−44.8 to −33.8)	<0.001	−1.1% (−7.8 to 6)	0.749	−4.1% (−11.1 to 3.5)	0.282	−28.8% (−35.1 to −22)	<0.001
Emergency	−34.1% (−41.2 to −26.2)	<0.001	−0.2% (−8.2 to 8.6)	0.966	−3.6% (−12.4 to 6.1)	0.455	−25% (−33.8 to −15)	<0.001
Priority A	−55.6% (−62.9 to −46.8)	<0.001	−7.7% (−19.9 to 6.5)	0.272	−8.1% (−20 to 5.6)	0.236	−33.2% (−42.6 to −22.3)	<0.001
Priority B	−17% (−37.0 to 9.5)	0.188	14.5% (−12 to 49)	0.315	8.5% (−16.9 to 41.7)	0.550	−31.7% (−49.7 to −7.2)	0.015
***Low Priority***	−60.3% (−64.2 to −56)	<0.001	−29.6% (−35.4 to −23.4)	<0.001	−17.6% (−23.6 to −11.1)	<0.001	−12.6% (−18.9 to −5.8)	<0.001
Priority C	−61% (−65.3 to −56.3)	<0.001	−31.4% (−37.6 to −24.6)	<0.001	−19.4% (−25.8 to −12.3)	<0.001	−13% (−19.9 to −5.5)	0.001
Priority D	−57.3% (−66.2 to −46)	<0.001	−20.8% (−35.1 to −3.4)	0.022	−8.8% (−23.8 to 9.3)	0.319	−10.7% (−25.6 to 7.2)	0.224
**Outpatient** **Visits**	−56.8% (−57.8 to −55.7)	<0.001	−7.1% (−8.9 to −5.3)	<0.001	−9.4% (−11 to −7.7)	<0.001	−9% (−10.7 to −7.2)	<0.001
**ED** **visits**	−65.2% (−66.3 to −64.1)	<0.001	−15% (−17.1 to −12.8)	<0.001	−30.3% (−32.1 to −28.5)	<0.001	−40.6% (−42.4 to −38.8)	<0.001

^1^ Comparison between the pandemic year (from 1 March 2020 to 28 February 2021) and the average numbers of the previous two years was performed using a mixed effects Poisson regression model with the center as the random effect. ED: Emergency Department.

## Data Availability

The data that support the findings of this study are available on request from the corresponding author (Giovanni Trisolino). The data are not publicly available due to them containing information that could compromise the privacy of research participants.

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
