# Peer review of "Resilience Against COVID-19: How Italy Faced the Pandemic in Pediatric Orthopedics and Traumatology"

_children, 2021, doi:10.3390/children8070530_

Round 1

Reviewer 1 Report

The authors attempt to quantify the known reduction in non-essential orthopedic surgery in seven centers in Italy. There is no question that case loads went down. This paper attempts to put numbers to this fact.

The priority classification scheme does not include traumatic injuries or emergencies. The most urgent (Priority A) is within 30 days. It appears the investigators included emergent cases in with Priority A and B, calling all three "high-priority". That lumps emergent cases in with those that could be delayed up to 90 days. This is somewhat of an odd mix. Not surprisingly, hospitals doing more trauma saw less decline in cases for 2020 at these can not be delayed. It may be better to separate trauma/emergencies from the other 4 categories.

Decline in trauma surgery seen in other countries is a surprise. Most US institutions have seen huge surges in all trauma volumes in 2020. Why is pediatric ortho trauma different? Might we worthy of comment/discussion in the paper. The impact of the paper is not really that it quantifies a KNOWN decline in surgery. The interesting part may be WHY certain types of cases decline more than others. Compare these data with Western penetrating trauma in adults - vastly different. 

Examples where an English editor may need to clean up the language a bit:
Line 69: "During this year, we have assisted to a drastic contracture of the so-called..."
Line 76: "...published specific recommendations, to pro- 76 crastinate elective or non-essential activities..."
Line 90: "The study involved seven public institutions, all of them were regional tertiary refer- 90 ral centers for pediatric orthopedics, included in the panel of the recommended hospitals 91 of the SITOP, that consists of fifteen hospitals."

Author Response

The authors attempt to quantify the known reduction in non-essential orthopedic surgery in seven centers in Italy. There is no question that case loads went down. This paper attempts to put numbers to this fact. The priority classification scheme does not include traumatic injuries or emergencies. The most urgent (Priority A) is within 30 days. It appears the investigators included emergent cases in with Priority A and B, calling all three "high-priority". That lumps emergent cases in with those that could be delayed up to 90 days. This is somewhat of an odd mix. Not surprisingly, hospitals doing more trauma saw less decline in cases for 2020 at these can not be delayed. It may be better to separate trauma/emergencies from the other 4 categories.

We agree with the reviewer, we separated trauma surgery from high-priority and low-priority surgery.

Decline in trauma surgery seen in other countries is a surprise. Most US institutions have seen huge surges in all trauma volumes in 2020. Why is pediatric ortho trauma different? Might we worthy of comment/discussion in the paper. The impact of the paper is not really that it quantifies a KNOWN decline in surgery. The interesting part may be WHY certain types of cases decline more than others. Compare these data with Western penetrating trauma in adults - vastly different.

REPLY:We thank the reviewer for this comment. We checked in the literature and found several studies that confirmed a decrease in trauma surgery both in children and adults, in Europe as well as in US (Sugand 2020, Bram 2020, Baxter 2020, ghafil 2021, Patel 2021, Berg 2021). Conversely, we were not able to find articles demonstrating an absolute increase in trauma surgery during the pandemic. Concerning the reasons why certain types of cases declined more than others, this is more difficult to explain. For instance, we observed that, while the overall volumes of trauma patients decreased during the pandemic, the ratio of cases requiring surgery increased. This phenomenon could be explained by the potential beneficial effects of some telemedicine program, as explained in the manuscript. Alternatively, Lock-down and quarantine related measures, such as school closings and social isolation could have increased domestic violence, child abuse or neglect and self-harm. Our study could not further investigate the impact of telemedicine rather than the effect of lockdown fatigue in changing injury pattern, since it did not distinguish the type, causes and severity of trauma requiring surgery, Therefore, we cannot provide further evidence in favor of or against these hypotheses. Nonetheless, we emphasize that the implementation of telemedicine along with outreach and injury prevention messaging may be useful in response to changing injury patterns. These considerations were added in the discussion.

Examples where an English editor may need to clean up the language a bit: Line 69: "During this year, we have assisted to a drastic contracture of the so-called..." Line 76: "...published specific recommendations, to pro- 76 crastinate elective or non-essential activities..." Line 90: "The study involved seven public institutions, all of them were regional tertiary refer- 90 ral centers for pediatric orthopedics, included in the panel of the recommended hospitals 91 of the SITOP, that consists of fifteen hospitals."

REPLY: Thank you. We edited the manuscript in a way to make it more clear.

Reviewer 2 Report

The authors made a cross sectional observational study titled: Resilience against COVID-19. How Italy faced the pandemic in pediatric orthopedics and traumatology.

-Firstly, I would like Authors to remove adverb "prospectively" from description of the study. If this is an observational multicenter cross-sectional study should be stated in that form. If the study is retrospective with one arm prospectively maintained then should be stated as such and an institutional review board should have given the permission to collect data.

- In the division of the hospitals in high volume and trauma-majority hospitals is there any overlaping – it would be interesting to see that – maybe as an chart or table? A suggestion of making a table with hospital names assigned to certain category.

-Table 1 should be text edited (font and alignment)

-What is Table 2S? A typing error or?

-In text there is and description of Table 4S and 5S even; I cannot see that tables in manuscript!

-I would suggest making and showing the results separately in division to southern and northern hospitals and additionally list the reasons by hospitals (lack of medical staff/lack of patients/lack of equipment...)

-Line 74; I suggest writing: “to face this state of emergency”

Author Response

The authors made a cross sectional observational study titled: Resilience against COVID-19. How Italy faced the pandemic in pediatric orthopedics and traumatology.

-Firstly, I would like Authors to remove adverb "prospectively" from description of the study. If this is an observational multicenter cross-sectional study should be stated in that form. If the study is retrospective with one arm prospectively mantained then should be stated as such and an institutional review board should have given the permission to collect data.

We agree with the reviewer. Despite the hospital databases are by nature prospective data collections, their investigation may be prospective or retrospective. In our case, the study was retrospective. We changed the text accordingly.

- In the division of the hospitals in high volume and trauma-majority hospitals is there any overlaping – it would be interesting to see that – maybe as an chart or table? A suggestion of making a table with hospital names assigned to certain category.

We thank the reviewer for this suggestion. We added a further table in supplementary material, indicating the type of hospital.

-Table 1 should be text edited (font and alignment)

OK

-What is Table 2S? A typing error or?

-In text there is and description of Table 4S and 5S even; I cannot see that tables in manuscript!

Tables 1S, 2S, 3S, 4S and 5S are reported in the supplementary material.

-I would suggest making and showing the results separately in division to southern and northern hospitals and additionally list the reasons by hospitals (lack of medical staff/lack of patients/lack of equipment...)

We thank the reviewer for this suggestion. We added a further table in the supplementary materials illustrating the situation in southern and northern hospitals, taken separately. Concerning the reasons by hospitals, all centers modified their activities to adapt to the pandemic. In particular, during the first wave almost all the elective activities were stopped in the northern centers since there was a dramatic outbreak of COVID-19, and the local health services had to face with an urgent need to save each resource to cope with the pandemic. During the following months elective activities were resumed, but with reduced numbers, to allow distancing among hospitalized patients, to reduce the crowding of the wards, to redistribute surgical rooms etc. Conversely, southern hospitals were less impacted by the first wave and the related restrictions, maintaining, occasionally even increasing, their surgical activities. A complete list of the possible reasons of the decline of medical and surgical activities by single hospital is, to say the least, arduous; several factors related to the hospital and the local health service organization, equipment availability, patient conditions, may have contributed in different ways to this phenomenon. Unfortunately, our study cannot investigate the effect of each condition on this decline. However, based on our findings, we can argue that the main factors affecting the drop and changes in pediatric  orthopedic activities were mainly related to the local burden of the pandemic, the effects of lockdown, the recommendations to postpone the majority of elective activities.

-Line 74; I suggest writing: “to face this state of emergency”

We changed the text according to your suggestion.